# Emerging Computational Approaches for Antimicrobial Peptide Discovery

**DOI:** 10.3390/antibiotics11070936

**Published:** 2022-07-13

**Authors:** Guillermin Agüero-Chapin, Deborah Galpert-Cañizares, Dany Domínguez-Pérez, Yovani Marrero-Ponce, Gisselle Pérez-Machado, Marta Teijeira, Agostinho Antunes

**Affiliations:** 1CIIMAR—Centro Interdisciplinar de Investigação Marinha e Ambiental, Universidade do Porto, Terminal de Cruzeiros do Porto de Leixões, Av. General Norton de Matos, s/n, 4450-208 Porto, Portugal; dany.perez@ciimar.up.pt; 2Departamento de Biologia, Faculdade de Ciências, Universidade do Porto, Rua do Campo Alegre, 4169-007 Porto, Portugal; 3Departamento de Ciencia de la Computación, Universidad Central Marta Abreu de Las Villas (UCLV), Santa Clara 54830, Cuba; deborah@uclv.edu.cu; 4Proquinorte, Unipessoal, Lda, Avenida 5 de Outubro, 124, 7º Piso, Avenidas Novas, 1050-061 Lisboa, Portugal; 5Universidad San Francisco de Quito (USFQ), Grupo de Medicina Molecular y Translacional (MeM&T), Colegio de Ciencias de la Salud (COCSA), Escuela de Medicina, Edificio de Especialidades Médicas and Instituto de Simulación Computacional (ISC-USFQ), Diego de Robles y vía Interoceánica, Quito 170157, Ecuador; ymarrero@usfq.edu.ec; 6EpiDisease S.L—Spin-Off of Centro de Investigación Biomédica en Red de Enfermedades Raras (CIBERER), 46980 Valencia, Spain; giselle.perez@epidisease.com; 7Departamento de Química Orgánica, Facultade de Química, Universidade de Vigo, 36310 Vigo, Spain; qomaca@uvigo.es; 8Instituto de Investigación Sanitaria Galicia Sur, Hospital Álvaro Cunqueiro, 36213 Vigo, Spain

**Keywords:** artificial intelligence, machine learning, AMPs, evolutionary algorithms, molecular descriptors, complex networks, proteogenomics

## Abstract

In the last two decades many reports have addressed the application of artificial intelligence (AI) in the search and design of antimicrobial peptides (AMPs). AI has been represented by machine learning (ML) algorithms that use sequence-based features for the discovery of new peptidic scaffolds with promising biological activity. From AI perspective, evolutionary algorithms have been also applied to the rational generation of peptide libraries aimed at the optimization/design of AMPs. However, the literature has scarcely dedicated to other emerging non-conventional in silico approaches for the search/design of such bioactive peptides. Thus, the first motivation here is to bring up some non-standard peptide features that have been used to build classical ML predictive models. Secondly, it is valuable to highlight emerging ML algorithms and alternative computational tools to predict/design AMPs as well as to explore their chemical space. Another point worthy of mention is the recent application of evolutionary algorithms that actually simulate sequence evolution to both the generation of diversity-oriented peptide libraries and the optimization of hit peptides. Last but not least, included here some new considerations in proteogenomic analyses currently incorporated into the computational workflow for unravelling AMPs in natural sources.

## 1. Introduction

The rise of resistance to antimicrobial agents evidenced in the last decades have caused excess healthcare costs worldwide [1]. The microbial natural resistance process, moved by evolutionary events, has been accelerated by the over-prescription and misuse of antibiotics [2]. This worrying situation has encouraged the search of new antibiotics from antimicrobial peptides (AMPs) with the ability to overcome resistance, mainly given by their versatile mode of action [3].Indeed, AMPs are not only considered for the development of antibiotics to treat multi-resistant bacterial strains [4,5], but also they are promising for the developing of antitumoral [6], antiviral [7], antifungal agents [8] and so on.

The discovery of peptides with relevant biological activities is a real challenge considering the great diversity of AMPs in terms of origin, structure, mode of action, activity, and, on the other hand by considering the overabundance of natural-occurring non-bioactive peptides [8]. Thus, several AMP databases with associated machine learning (ML)-based classifiers have been developed for over one decade, in order to assist wet-lab researchers in the long development process of peptide-based drugs [9]. AMP databases such as DAMPD [10], CAMPR3 [11], LAMP [12], DRAMP [13], ADAM [14], DBAASP [15] have incorporated ML predictors trained with alignment-free (AF) protein features such as amino acid (aa) and pseudo-aa composition, structural features, word frequency-based features, physicochemical aa properties with influence on the AMP activity, and some others [16,17] (Table 1). Figure 1 illustrates how databases and ML algorithms have been integrated to assist the discovery/design of AMPs for the developing of peptide drugs. 

The prediction tools built up with Support Vector Machine (SVM) and Random Forest (RF) based classifiers have been widely applied, but hardly considered the natural imbalance between the AMPs and non-AMPs [18]. On the other hand, emerging ML techniques such as Deep-Learning Neural Networks [18,19,20,21] and those based on the Rough Set Theory [22,23] have been applied to improve certain classification pitfalls like the quality in the learning phase and the classification boundaries between AMPs and non-AMPs, respectively. Although most of the classical predictive tools have focused on if a query peptide is an AMP or not, without targeting a specific biological activity among the reported for the AMPs [24], the current tendency is to address a hierarchical multi-level classification by downstream considering the specific biological activities of the AMPs as labels e.g., the antibacterial, antifungal, antiviral and antitumoral among others.

The most popular hierarchical multi-label classifiers, also listed in Table 1, are the following: (i) the iAMP-2L, a two-level classifier trained with Chou’s pseudo amino acid composition (PseACC) [25], aimed at identifying AMPs and their five functional types [26], (ii) the iAMPpred predictor that combines compositional, physicochemical, and structural features into Chou’s general PseACC for training a SVM multi-classifier [16], (iii) the MLAMP, a RF-based classifier built up with a non-classical PseACC sequence formulation incorporating a Grey Model that firstly discriminates AMP from non-AMPs, and then subclassify their biological activities into antibacterial, anti-cancer, antifungal, antiviral, and anti-HIV [27], (iv) the Antimicrobial Activity Predictor (AMAP) [28], a hierarchical multi-label classifier targeting 14 biological activities that is built up with SVM and XGboost tree [29] algorithms trained with amino acid composition (ACC) features, (v) the AMPfun webserver containing RF-based models that firstly classify AMPs and non-AMPs and afterwards address the prediction of AMPs functional activities including their possible target types [30], and more recently, the (vi) AMPDiscover [31] and the (vii) ABPFinder webservers (https://protdcal.zmb.uni-due.de/ABP-Finder/index.php; accessed on 7 March 2022) containing hierarchical RF-based classifiers built up with protein descriptors from the ProtDCal software [32] to firstly detect AMPs and antibacterial peptides (ABPs), respectively. While the AMPDiscover uses several downstream RF models to predict AMPs specific functions (antibacterial, antifungal, antiparasitic and antiviral), the ABP-Finder sub-classifies ABPs according to the Gram staining type of the potential targets (Gram-positive, Gram-negative bacteria, or broad-spectrum peptides with expected activity against both types of bacteria) by using a multi-classifier. The high success classification rates of both tools stems from considering the StarPep database [33] which is probably the most comprehensive curated repository of AMPs so far, and from performing an applicability domain (AD) analysis for the proposed ML models [31]. Both, AMPDiscover and ABP-Finder defined ADs for their corresponding RF-based models, however, AMPDiscover perform a rigours AD analysis at applying a consensus-based decision from five different approaches [31].

Despite the great number of reported ML-based tools for AMPs prediction, only few ones have considered the lack of balance among either the specific activities of AMPs or among their putative targets, as well as the AD of their corresponding models. The imbalance among AMPs and non-AMPs as well as the existing one among AMP activities was addressed by applying the synthetic minority over-sampling technique (SMOTE) during the IAMPE and MLAMP building [27,34] while the ABP-Finder addressed the imbalance among the bacterial target types of the ABPs (Gram+, Gram- and Gram+/- bacteria) by training a RF multi-classifier with a cost matrix weighting the different types of misclassified cases according to the imbalance ratio between the two classes (https://protdcal.zmb.uni-due.de/ABP-Finder/index.php; accessed on 7 March 2022).

On the other side, artificial intelligence (AI)-derived approaches like evolutionary algorithms have been applied to optimize lead candidates retrieved from the high-throughput screening in drug discovery. Evolutionary algorithms are inspired on several evolutionary events occurring in nature; they generally start with a small population of peptides identified as putative leads due to its relevant biological activities. The optimization is carried out by the generation of offspring peptides from these initial peptides by applying several operators simulating natural evolutionary process like cross-over and mutation operators, a parent and survival selection algorithms [40,41]. A parent selection algorithm is firstly applied on the initial peptide population to select the best parent peptides for the offspring generation. The survival aims at selecting a subset of good individuals (new population) from the generated offspring peptides. Then, the new peptide population will be iteratively subjected to the parent selection algorithm, evolutionary operators and the survival selection until finding an offspring peptide meeting a termination condition (selection criteria in Figure 2). The selection criteria can be represented by a fitness function which can be a ML model scoring peptide bioactivity. This selection process may be accompanied to experimental evaluations against the desired biological activities [42] (Figure 2).

The genetic algorithm is the most popular technique among the evolutionary approaches where the peptides with promising biological properties (initial solution) are encoded as binary strings into chromosome-like structures, called genotypes. The optimization process is performed by evolving each chromosome toward optimized solutions by iteratively applying genetic recombination (crossover) operators and survival fitness functions that is somehow similar to the parent selection mechanism [40,42,43,44]. Optimized solutions in the case of peptides consist in generating structural entities with optimized biological properties e.g., peptides showing a trade-off among their pharmaceutical potency, solubility, haemolytic and toxicity properties [42] (Figure 2).

Despite AI-derived approaches have been largely applied to the rational search and design of bioactive peptides; most of them are represented by classical ML and evolutionary algorithms that frequently also use canonical sequence-based features as peptide descriptors and therefore have been documented in literature [18,45,46]. However, there is a growing number of emerging computational approaches effectively applied to the search/design of bioactive peptides that are comprehensively revisited here (Table 1).

Most of the non-standard approaches are represented by classical ML algorithms which are either trained with non-conventional peptide features [31] or combined with sequence alignment methods [47]. In addition to the singularity of these predictors; pre-processing steps managing the natural imbalance between bioactive and inactive peptides have been hardly applied to the AMPs predictions [27,34] as well as no big data solutions have been implemented yet to address scalability problems. As mentioned before, other less-known ML algorithms in the field of protein/peptide science like those based on the Rough Sets Theory (RST) are being currently intended for peptide classification/design [22,48]. Moreover, a non-conventional methodology that analyses the known chemical space of bioactive peptides by similarity networks was developed to identify the most relevant ones for each specific biological activity [33]. Such representative peptides were recently used in multi-query similarity searches against the StarPep database to repurpose AMPs for specific activities such as antiparasitic and tumour homing [49,50].

By other side, evolutionary algorithms that simulate sequence evolution have been recently applied to design/optimize peptides having a pharmaceutical activity [51,52]. Last but not least, computational tools used in proteogenomic analyses are being modified for uncovering cryptic peptides with biological activities in natural sources [53,54]. From now on we go deeper into these emerging approaches in peptide search and design

## 2. Non-Classical Peptide Features for Bioactivity Prediction

### 2.1. Peptide Features Inspired in Molecular Descriptors Used in Cheminformatics

There is a set of chemoinformatics-derived peptide features considered as “non-conventional” because of its in-house development; however, have been successfully applied in the recognition of bioactive peptides by ML-based classifiers [31,55,56,57,58]. The definition of these peptide/protein features is generally inspired on the mathematical formalisms applied to the calculation of molecular descriptors for small organic molecules [59,60], which have been traditionally used to Quantitative-Structure-Activity Relationship (QSAR) studies for drug design/search. Most of them are classified as topological descriptors since they consider the connectivity either between adjacent amino acids (aas) or between aa groups by using both algebraic and statistic invariants [32,61,62].

Those based on algebraic forms express protein/peptide structural topology through the definition of connectivity or adjacency matrices. The elements of these matrices (n_ij_ or e_ij_) reflect topological relationships between the aas or aa groups, they are equal to 1 if i and j are adjacent otherwise take the value of 0. Topological indices (TIs) are estimated by applying several algorithms on the connectivity/adjacency matrix. The most common algorithms for the TIs calculation involve the powers of the topological matrix, the multiplication of a property vector by the topological matrix and the multiplication of vector-matrix-vector (Figure 3). Many of the most popular TIs within the cheminformatic have been defined by these algebraic formalisms, such as the Winner index (W) [63], the Randić invariant (χ) [64], Broto–Moreau autocorrelation (ATSd) [65], the Balaban index (J) [66], and the spectral moments introduced by Estrada [59]. Thus, many of them were reformulated to describe the spatial topology of aa sequences at different structural levels, e.g., linear sequences (1D), pseudo-secondary structure (2D) and the 3D-dimensional space [61,62] (Figure 3).

On the other hand, there is another set of topological descriptors that also comes from the chemoinformatic field that have been applied to the identification and design of AMPs [31,52,56,58]. They are not formulated by using algebraic forms but rather they rely on descriptive statistics as invariant operators on the aa properties either along the sequence or the 3D protein structure. In this case, the 1D or 3D topology is encoded by the application of classic cheminformatics algorithms that consider the neighborhood such as autocorrelation [65], Kier-Hall’s electro-topological state [68], Ivanshiuc-Balaban [69], and Gravitational-like operators [70].

#### 2.1.1. Topological Indices from Algebraic Forms

Among the TIs defined for small molecules, the spectral moments formalism probably is one of the most extended to characterize proteins and peptides structures [61,62,71,72]. The spectral moments may encode peptide structures through the definition of their corresponding topological matrixes and the application of the trace operator on the k-th power of such matrixes (Figure 3).

A sort of stochastic spectral moments applied to the electronic or charge delocalization of the aas within the peptide backbone and the entropy involved on such delocalization, were applied to model the bitter tasting threshold of dipeptides by linear discriminant and regression analyses [57]. These non-standard peptide features provided accuracies higher than 83% in the detection of bitter taste, and the regression models could explain the experimental variance of the bitter tasting threshold in more than 80%. It was shown the non-standard peptide descriptors correlate with the bitter taste as good as or even better than other well-known peptide features like the z-scale [73].

The spectral moments have been also applied to characterize bacteriocins. Bacteriocins are peptidic toxins produced and exported by bacteria as a defense mechanism to kill or inhibit the grow of other strains but the producer. The bacteriocins are very attractive for the development of new antibiotics and anticancer agents, however their high structural diversity represents a challenge for alignment-based predictive tools. Since the hydrophobicity and basicity of bacteriocins are relevant for their antibacterial activity, Agüero-Chapin et al. introduced the 2D-Hydrophobicity and Polarity (2D-HP) maps to pseudo-fold bacteriocin protein sequences in order to derive a set of spectral moments encoding information beyond the linear sequence [74] (Figure 4). These TIs are implemented in the Topological Indices to Biopolymers (TI2BioP) software [75] and were useful to build an AF model based on Linear Discriminant Analysis with a higher sensitivity (66.7%) than the attained by InterProScan (60.2%). In addition, they could detect cryptic bacteriocins, ignored by alignment methods [74].

#### 2.1.2. Topological Indices from Descriptive Statistics

The cheminformatic-derived protein descriptors that have been widely applied to the prediction and design of bioactive peptides were developed and implemented by Ruiz-Blanco et al. in the ProtDCal software [32]. ProtDCal provides a great diversity of protein/peptide descriptors thanks to its divide-and-conquer methodology that considers both the aa properties and those estimated for groups, which can be modified by the neighbourhood through the application of classic previously-mentioned chemoinformatics algorithms. The modified properties of the aas or their resulting groups are later aggregated using statistical operators to estimate local or global descriptors either at sequence or 3D structural level. Although a more detailed description of ProtDCal’s protein descriptors can be found in [32], the Figure 5 shows an schematic representation of the protein descriptor generation process of ProtDCal. The diversity of ProtDCal’s protein descriptors represented by different families stems from combinatorically applying different aa properties, the ways to consider the vicinity to the target aa by several operators, the criteria used to group the aas as well as the invariant operator used for aggregating aa properties within the same array (Figure 5).

ProtDCal’s descriptors have been involved in the discovery of antibacterial peptides by developing a non-conventional multi-target QSAR models [56]. Despite the AMPs selected for training were evaluated against multiple targets (Gram-positive bacterial strains), they could be integrated in the same model by modifying their ProtDCal’s descriptors through the Box-Jenkins moving average operator. This operator allows modifying the sequence-based descriptors by subtracting the corresponding mean of the descriptors of all AMPs assayed against the same Gram-positive bacterial strain. This is a way to particularize a sequence-based descriptor by incorporating information about the experimental conditions or biological assays. With this kind of descriptors, the multi-target cheminformatic model displayed percentages of correct classification higher than 90.0% in both training and prediction (test) sets [56].

Similarly, the same authors also applied the Box-Jenkins moving average operator to develop non-conventional multi-task QSAR models able to predict simultaneously antibacterial activity and toxicity [58]. This time, the continuous response variables measured on AMPs such as minimum inhibitory concentration (MIC), cytotoxic concentration at 50% (CC50), and haemolytic concentration at 50% (HC50) were transformed in a binary variable labelled as (1) referred to high antibacterial activity/low cytotoxicity, and (−1) assigned to low antibacterial activity/high cytotoxicity. The ProtDCal’s descriptors that usually encodes only peptide features were modified by the Box-Jenkins moving average operator in order to consider the variability implying the evaluation of the antimicrobial activity and toxicity on different biological systems. Thus, a multi-task QSAR model displayed an accuracy higher than 96% for classifying/predicting peptides was built by using LDA discriminant [58].

ProtDCal’s descriptors have also been involved in the design of new peptides that inhibit the *E. coli* ATP synthase, as putative antibiotics [52,76]. ProtDCal’s descriptors, implemented in PPI-Detect [77], were applied to predict interactions between peptides and the main subunits of *E. coli*’s (Ec) and human’s (Hs) F1Fo-ATP synthase. Those peptide with a maximum and a minimum interaction likelihood with EcF1Fo and HsF1Fo were selected for in vitro assays. An overall of three peptides resulted attractive for further optimization steps in the design of new antibiotics [52,76].

More recently, ProtDCal’s protein descriptors were successfully applied to improve the prediction performance of the existing alignment-free models by using the largest experimentally validated non-redundant peptide dataset reported to date, the StarPepDB [78], together with Random Forest (RF) classifiers [31]. Pinacho-Castellanos at al. not only built RF-based models for identifying AMPs, but also addressed the main biological activities reported for them (antibacterial, antifungal, antiparasitic, and antiviral) as endpoints. The specific functions of AMPs were either directly predicted or by a hierarchical classification that first consider the antimicrobial activity. RF-based models, developed with ProtDCal’s descriptors aimed to predict specific activities of AMPs, showed a higher effectivity and reliability than 13 freely available prediction tools. The best reported models were implemented in the AMPDiscover tool [31], publicly available at https://biocom-ampdiscover.cicese.mx/ (accessed on 7 March 2022). Ruiz-Blanco et al. also applied successfully ProtDCal’s descriptors to predict antibacterial peptides by using RF-based models trained with StarPepDB instances and, in a second step they are predicted on what bacterial targets according to their Gram-staining classification could be active by using a multi-classifier. These two RF-based models were implemented in the web server ABP-Finder: https://protdcal.zmb.uni-due.de/ABP-Finder/ (accessed on 7 March 2022) which is freely available but unpublished yet.

### 2.2. Integration of Peptide Features from Heterogenous Sources

Considering previous experiences in protein functional classification where protein features from heterogeneous sources have been integrated to improve classification rates; we wonder if this strategy has been applied to peptide classification? In this sense, the integration/combination of alignment-based (AB) and alignment-free (AF) protein features in machine learning models have been evaluated for such purpose. For example, Galpert et al. improved orthologs classification at the twilight zone (<30% of identity) by combining AB and AF protein similarity measures in supervised big data classifiers [79]. It has also been shown that the integration of AB and AF methods gives the best exploration of highly diverse protein classes, such as the nonribosomal peptide synthases (NRPS) represented by their A-domains [80]. Other examples of feature integration methods for remote homology detection can be found in [81], and the one of Borozan’s et al. [82], based on weighted aggregation which is a very inclusive approach avoiding the loss of information.

Regarding AMPs classification improvements by integrating AB and AF peptide features, an algorithm applying AB measures and the SVM algorithm trained with AF pairwise measures was published for increasing AMPs prediction sensitivity [47]. The algorithm consists in two stages. Firstly, AMPs are identified by Basic Local Alignment Search Tool (BLAST) scores, and those peptides that cannot be unequivocally identified by pairwise alignments were inputted in an SVM-based classifier built with AF pairwise similarity scores. The AF similarity scores were estimated with the Lempel–Ziv’s complexity algorithm [83]. The integrative algorithm achieved higher sensitivity performance for AMPs prediction than the prediction tools implemented within the first version of CAMPR3 database [11] and the integrated method proposed by Wang et al. [84]. Wang and colleagues had previously proposed a similar algorithmic workflow where BLAST is used to firstly classify a query peptide against a training set made up by 870 AMPs and 8661 non-AMPs. Classification label is transferred to the query peptide from the matching with highest similarity score. Query peptides that did not match with any within the training set were encoded by protein features like ACC and PseACC and the aas by five of their physicochemical and biochemical properties. As the number of generated features were relatively high, a rigorous feature selection step was performed by applying both the Maximum Relevance, Minimum Redundancy (mRMR) method [85] and the Incremental Feature Selection method [86] before building a Nearest Neighbour (NN)-based predictor. The NN algorithm assign the label AMP or non-AMP to a query peptide according to the class of the nearest neighbour.

Despite the efforts for integrating AB and AF features in a classification peptide system; they have actually been combined through their corresponding algorithms and have not been included in the same model or function. In this sense, AB and AF similarity scores could be combined to build an unique classifier for AMP prediction, as Galpert et al. did it for ortholog detection [79].

### 2.3. NMR-Based Features for Peptides

In 2020, the IAMPE webserver (http://cbb1.ut.ac.ir/; accesed on 17 March 2022) was released for an accurate prediction of AMPs by using classical ML-based classifiers trained with both conventional and ^13^CNMR-based features. The non-conventional ^13^CNMR-based features for peptides were defined from the quantitative NMR spectra for ^13^C isotope of the naturally-occurring aas. Firstly, ^13^CNMR-based features for each aa were calculated using ^13^CNMR spectra signals. Secondly the aas were grouped according to their ^13^CNMR-based features by applying Fuzzy c-means clustering algorithm. The resulting aa clusters were used to extract feature vectors along the peptide sequences according to classical “composition”, “transition” and “distribution” patterns. Despite the new information provided by such non-conventional peptide descriptors, authors suggested their combination with physicochemical features to yield higher accuracy for the prediction of active AMP sequences [34].

## 3. Breakthroughs of ML Algorithms in the AMP Prediction

### 3.1. Data Imbalance and Multi-Label Classification in the Prediction of AMPs—New Algorithm Approaches

As mentioned in the Introduction, data imbalance is an issue to tackle in the classification of potential peptide sequences. Here, we collected some other reported solutions combining two-level classifiers with imbalance management in both, the first level binary AMP/non-AMP problem, and the second level multi-label functional type problem. For example, the authors of MAMP-Pred [87] proposed two alternative imbalance management methods: (i) under-sampling of the non-AMP class, and (ii) weighting sequences according to the imbalance ratio; the second one being eligible after the experiment process. Then, they used pruned sets and label combinations, considering label correlations, to transform the RF binary classification. For the classification assessment, the Matthew’s correlation coefficient was selected for the first level, and the multi-label metrics: Exact-Match Ratio (EMR), Hamming-Loss (H-Loss), Accuracy (Acc), Precision (Precision, Recall), Ranking-Loss (RL), Log-Loss, One-error (OE), F1-Measure (F1-Mic, F1-Mac), for the second level. As they assessed, MAMP-Pred outperformed iAMP-2L (proposed in 2013 as a two-level multi-label classifier) because of the feature extraction process involved ACC and its eight physicochemical selected properties, besides the classification process.

Another example of imbalance management can be found in [88] where the authors tried to identify peptides with dedicated anti-CoV antimicrobial function on an imbalanced dataset with relatively insufficient positive data. They used NearMiss under-sampling and balanced RF to build the classification model, and the sensitivity, specificity and geometric mean for the unbiased evaluation.

Ensemble learning has also been used to cope with class imbalance in the binary AMP/non-AMP prediction tool Ensemble-AMPPred [89]. The prediction model based on ensemble methods (RF, max probability voting, majority voting, adaptive boosting, or extreme gradient boosting) was combined with feature extraction (vectors of 517 numerical descriptors representing peptide sequences), feature engineering (hybrid feature generation by the fusion of various selected features using a logistic regression model) and feature selection to improve classification accuracy after the application of a balancing clustering-based proportionate stratified random sampling that selected peptide sequences representing the positive and negative data. Thus, representative sequences selected from each cluster were used as training data, while the other remaining sequences, as testing data.

A recent report in [90] presents a multi-label framework HMD-AMP to hierarchically annotate peptide sequences into AMP/non-AMP, and then, into eleven functional classes that can be small and extremely imbalanced classes. The classification framework includes an embedding layer of protein sequences, a protein language encoder, a feature transformer and a hierarchical deep forest model. An ablation study and a reduced feature test demonstrate the effectiveness of the framework based on the detailed structural information of AMPs to improve the accuracy of the prediction model and to manage data imbalance problem. At each function prediction level, the model demonstrates a cascade forest structure where each cascade level is an ensemble of decision tree forests, and different types of forests are included to make the model diverse. It’s worth noting that deep forest does not rely on backpropagation, so it is suitable for training data with either imbalance labels or small sample sizes, hence preventing the model from overfitting.

### 3.2. Deep-Learning in the Recognition of AMPs

The lack of samples in the positive class, as well as, the ambiguity in the negative class are key issues concerning deep learning models in AMP prediction as stated in review [91]. The starting point for knowledge discovery in this rough scenario is the correct representation of raw data. Precisely, deep learning provides a solution to the human expert dependence problem of featurization, which is known as representation learning; but also allows the application of some widely-used features in peptide machine learning by means of unsupervised embeddings (pretrained representations that can be fine-tuned with specific downstream supervised tasks), learned embeddings (usually one-hot or one-letter encoding on the amino-acid level, producing a dimension-reduced dense vector for subsequent layers), or engineered features (physicochemical or evolution-based properties).

In generative approaches for AMP discovery, recently reviewed in [92], the reliance on expertise-engineered features may limit the generation of candidates qualitatively distinct from known AMPs, or the limited number of known structures of the annotated peptides may reduce the effectiveness of structured-based models [93]. On the contrary, those attribute-controlled models based on recurrent neural networks, variational autoencoders, adversarial autoencoders, generative adversarial networks may encourage novelty of designed sequences. That is the case of the specific bidirectional conditional generative adversarial network developed in AMPGAN v2 [94] that learns data driven priors through generator-discriminator dynamics and controls generation using conditioning variables. Thus, a learned encoder mapping data samples into the latent space of the generator implements the bidirectional component that aids iterative manipulation of novel, diverse, and application-tailored candidate peptides.

The diversity target in generative models has been also tackled with a semi-supervised learning approach combined with a variational autoencoder (VAE) that can simultaneously learn from the large unlabelled peptide sequence databases and a limited number of labelled sequences as in PepCVAE [95]. In this case, a controlled generative model is learned from large unlabelled peptide database for the encoder and decoder losses, together with a much smaller labelled dataset (peptides with reported antimicrobial annotation) for the classifier loss, that is, using a large unlabelled corpus to capture the distribution with VAE, and a small labelled corpus to learn a certain controlling attribute code.

Also with VAE generation, the report in [96] used the Giant Repository of AMP Activity (GRAMPA) [97] to apply an improved automated semi-supervised approach based on stochastic long short-term memory (LSTM) encoder-decoder networks for generating promising new sequences and an experimental investigation, resulting in low minimal inhibitory concentration (MIC) AMPs against *Escherichia coli*, *Staphylococcus aureus*, and *Pseudomonas aeruginosa*. In this approach, the decoding from the same point in the latent space may result in a different peptide being generated and is dependent on the random seed set prior to running. Thus, the VAE is trained on a curated AMP dataset followed by the development of a regression model for activity prediction and the subsequent development of the latent space. Then, new AMP sequences are identified from the latent space (by sampling) and, subsequently, the AMPs are produced and characterized with their corresponding MIC values. This method produces peptides with similar MICs as the input reference peptides, but with novel sequences not found in the training set; at the same time, without imposing thresholds on peptide characteristics or otherwise biasing output post-sequence generation. As a result, a list of newly generated active peptides includes non-canonical AMPs of low helicity and low net charge.

An alternative data augmentation method is presented in [98] to improve the recognition of neurotoxic peptides via a convolutional neural network model. Novel potential neurotoxic peptides were discovered from the best performed model in a simulation dataset among the transcriptome of an endemic spider of South Korea, *Callobius koreanus* (*C. koreanus*). The BLAST-based augmentation method was intended to improve the generalization property of the model.

Specifically, for candidate short peptide generation, the authors in [99] combined LSTM generation and bidirectional LSTM classification to design short novel AMP sequences with potential antibacterial activity against *E. coli*. The models were trained using sequences with proven low MICs and tuned with Bayesian hyperparameter optimization.

Some other deep learning methods are reviewed in [100] as a promising approach to meet short-length peptides requirements [101] where they combine deep convolutional neural network with reduced aa composition comprising clustered aas on the basis of evolutionary information, substitution score, hydrophobicity, and contact potential energy. As a result, a short peptide of 20 aa was selected by Deep-AmPEP30 from sequences extracted from the gut commensal fungus *C. glabrata* genome and experimentally validated to have antibacterial activities similar to ampicillin.

In a recent review [39], the authors presented some reasons to select ML approaches over deep learning ones in AMP prediction and design, when a fair balance is required among high accuracy and generalization capability, interpretability and low computational cost. However, some improvements like parameter tuning or model hybridization may lead to more robust deep learning classifiers in this field.

### 3.3. Rough Sets Theory in the Classification of AMPs

As an example of model hybridization, the authors in [48] presented a codon-based genetic algorithm combined with rough set theory methods to find a peptide active against *S. epidermidis*. Their rough set theory method provided explicit boundaries between physicochemical properties that active sequences possess and inactive sequences do not possess. Since this method produced explicit decision components, they could test sequences containing multiple components. They were inspired in their previous publication [22] where they tried to reduce false discovery rate with a rough set-based classification method generating similarity rule set boundaries between active and non-active peptides based on their physicochemical properties.

Another example of the rough set theory application can be found in [102] where they implemented a rough set classification framework together with a Rough Set Quick Reduct and Rough Set Relative Reduct based on an improved Harmony Search algorithm to classifyAnti-HIV-1 peptides. Specifically, they hybridized a rough set-based feature selection technique, with population-based meta-heuristic algorithms (Particle Swarm Optimization), to classify the peptide sequences and solve dimensionality problems. Besides, a fuzzy set classification framework [23] was also intended to cope with limited and severely skewed high-dimensional space for short (<30 aa) AMP activity prediction.

## 4. Other Methodologies Than Classical ML for Identifying and Modelling AMPs

### 4.1. Homology-Based Prediction and Modelling of AMPs

The most popular approaches in addition to classical machine learning algorithms for the identification of AMPs in databases are local alignments which are represented by BLAST and FASTA tools [103,104]. Although local alignments have been successfully applied by using iterative rounds and filters such as the presence of signal peptides, aa patterns and gene vicinity during AMP searches [105,106,107], they can fail in identifying some AMP sequences [55], if compared to pattern-matching searches [107,108]. There are two main ways for searching for sequences by patterns: hidden markov models (profile-HMM) [109] or regular expressions (REGEX) [110]. Both the REGEX and profile-HMM methodologies work similarly for the identification of AMPs. Firstly, a set of homologous sequences are aligned and the multiple sequence alignment (MSA) is inputted to a specific program such as Pratt [111] or HMMER [112] for the identification of REGEX patterns or profile-HMM, respectively. Currently, instead of building REGEX patterns and profile-HMMs, they are available for many protein families at the Prosite [113] and Pfam [114] databases where a query sequence/peptide can be identified. The pattern/profile-based searches for AMPs can be complemented with the identification of signal peptides and other structural filters. In fact, improved versions of databases have incorporated MSA, profiles-HMM and molecular modelling for AMPs detection [11,115,116]. Even so, when a query peptide could be high-scored against profile-HMMs from different peptide families, it is advisable to use a prediction tool combining different protein signature recognition methods such as InterProScan [117].

As we previously mentioned, the molecular modelling complements AMPs pattern-based searches by confirming expected three-dimensional (3D) structural features characterizing them. The 3D structure can be also integrated into homology-based searches to identify homologous sequences sharing low identity but retaining a great structural conservation. Such structural similarities have enabled the detection of AMPs in databases with higher accuracy [9]. When the structures of peptides are not experimentally elucidated, two modelling techniques are suggested: homology-based and *ab initio* modelling. The homology-based modelling uses the structure experimentally-determined from available homologous as template to infer the 3D structure of novel peptides, but rather using structural than sequence similarities, especially if the query and template are remote homologous [118]. By contrast, the *ab initio* method is used to predict the structures of peptides with yet unknown homologs. The prediction of the 3D protein structure starts from scratch requiring an energy model describing the main factors that contribute to the stability of the folding process and an efficient method for the conformational space exploration of the peptide chain [119]. However, homology-based approaches are more suitable for peptides when homologs are identified. In fact, the second release of BACTIBASE [115] incorporated the MODELLER program [120], as a tool for the 3D structure prediction of query peptides by homology to known bacteriocins [115]. Besides, the incorporation of 3D structure prediction tools to AMP databases provide another filter for an accurate identification of query AMPs, the 3D structure can be used for scoring peptide-cellular target interactions which is a crucial step for the *in-silico* design of novel AMPs [121].

Especially, since classical ML algorithms were recently reviewed in [18], we have addressed here, traditional homology-based approaches applied to the search and the modelling of AMPs, and will describe next, the most singular algorithms.

### 4.2. Emerging ML-Independent Methodologies for AMP Prediction/Design

In this section, we will address other emerging methodologies regardless of ML approaches and classical homology-based approaches for AMP discovery. Firstly, we want to highlight the AMPA webserver (http://tcoffee.crg.cat/apps/ampa, accessed on 7 March 2022), developed to detect antimicrobial stretches within the protein sequences. The antimicrobial regions detected in proteins can serve as new templates for AMP design, especially those uncovered within proteins no related with the defense function. AMPA algorithm does not depend on homology-based searches since it estimates an antimicrobial index (AI) to each aa, derived from half-maximal inhibitory concentration (IC_50_) values in high-throughput screening experiments, encoding the propensity of each aa to be present in an AMP sequence. As low IC_50_ values correspond to high activity, aas with low AIs are more likely to be part of an AMP. By applying a sliding-windows analysis along the protein sequence, AMPA generates an antimicrobial profile based on the AIs. Those regions scored below certain threshold are considered putative antimicrobial domains [122]. The singularity of this approach is that it doesn’t either rely on building machine learning models or similarity searches against AMP databases. However, potentially conserved antimicrobial regions can be checked in conjunction with the T-coffee alignment tool [123].

On the other hand, complex networks have been applied to explore the chemical space of AMPs aimed to discover structural entities with promising biological activities that also could serve as template for peptide drugs design/optimization. In this sense, Marrero-Ponce et al. were the pioneers on this topic by publishing a seminal of related works [33,78,124]. Firstly, Marrero-Ponce et al. analyzed both the diversity among 25 AMP databases and the showed within each one. The study revealed some AMP databases contained common sequences showing certain overlapping degree. After removing duplicates among AMP databases, a representative set of 16 990 non-redundant AMPs was collected, which probably was the most comprehensive and exhaustively curated AMP dataset at that moment [124]. This relevant dataset was further enriched and structured in a graph database called StarPepDB (http://mobiosd-hub.com/starpep/; accessed on 17 March 2022) integrating 45 120 unique peptide sequences from 42 AMPs databases (Figure 6), with their metadata (origin organisms, function, biological target, source database, chemical modifications, cross-referenced entries to UniProt, PDB and PubMed) [78].

StarPepDB has a star-like network architecture where a central node represents the peptide sequence and is connected to neighbour nodes labelled with the metadata. The edges depict a relational and unidirectional connection of the central node by a using a set of selection criteria “produced by”, “assessed against”, “*related to*”, “*compiled in*” with its corresponding metadata nodes such as the origin, target, function and database, respectively. Peptide nodes besides the sequence also contain peptide’s ID and length, while the metadata nodes have the ‘name’ property and relationships have the ‘db-ref’ property (referred as source database) [78]. Finally, different network topologies can be visualized by applying filtering criteria on StarPepDB. For example, it is possible to display a network of those peptides (central nodes) “*related to*” (edges) function “antibacterial” (metadata node) and “*compiled in”* (edges) the ADP database (metadata node).

Thus, the StarPepDB structure together with the StarPep toolbox allows building customized networks and their visualization. The visual and analytics exploration of the network by extracting some centralities measures (e.g., weighted degree or harmonic centralities) allows identifying the most relevant bioactive peptides in the network (Figure 7). Furthermore, peptide subsets can be either retrieved from the graph database by sequence identity searches or by applying filtering criteria such as peptide length, sequence motifs/patterns, physicochemical properties, and other metadata.

More recently, the same research group encoded each peptide sequence with a set of molecular descriptors bearing non-redundant structural information to set alignment-free (AF) pairwise similarity/distance relationships among the peptide nodes of the network by using a general pipeline as show in Figure 7. The resulting chemical space represented by these AF similarity networks are explored by visual inspection in combination with clustering and network science techniques [49,50].

Here, we show the chemical space network (CSN) of 174 non-redundant Anti-Biofilm Peptides (ABPs) (Figure 8) by applying the StarPep Toolbox flowchart represented above. Networks become more interpretable through visual inspection if having a community structure. Note that communities of ABPs may represent some biologically relevant regions from the chemical space where bioactive compounds reside. Hence, we have explored the CSNs by varying the similarity threshold until a well-defined community structure emerged. In this way, a final CSN has been analyzed by adjusting the similarity threshold to 0.65, at network density of 0.0068, achieving 20 ABP outliers (singletons) with atypical or unique sequences (Figure 8). Also, for each peptide discovered to be a relevant node, additional information (metadata) is available in Appendix A.

Once a community structure is found, we rank nodes in decreasing order according to the community Harmonic centrality measure for retaining the top-*k* of the ranked list. Particularly, the top 10 exposes densely connected groups of nodes like cliques, which are defined to be complete subgraphs. These related sequences may be forming families in the chemical space of ABPs. These central peptides within each local leading community are given in SI1-B, and they may be representing sequence fragments or naturally occurring peptides that could be identified as starting structures for lead discovery. For instance, the peptide starpep_00000, starpep_05561, starpep_00361 are the most central nodes of the CSN (all in cluster 10). ABPs starpep_03668, starpep_04267, starpep_00004 and starpep_07895, starpep_12531, starpep_012529 are more central inside Communities 3 and 12, respectively (Figure 8 and SI1-C).

As can be observed in Table in SI1-C, some neighbor nodes within the communities may be representing a family of similar ABPs. Another example of closely related sequences can be seen in the 3 members of the Cluster 3 (see all ABPs in Community 3 in SI1-B). The peptides inside this cluster have the same length of 12 aas. So, it is expected that there are many ABPs with similar centrality values in the CSN, and it is advisable to extract some non-redundant ABPs from communities than just selecting the highest-ranked ones. To clearly extract central but non-redundant ABPs from each cluster (scaffold extraction, see Figure 7), we sort ABPs according to the decreasing order of their harmonic values. Then, the redundant sequences are removed at a given % of sequence identity. We have used an identity cutoff of > 35% to consider that a particular sequence is related to already-selected central ABPs and, as a consequence, removed from the CSN. Finally, the non-redundant 44 ABPs were ranked according to their decreasing values of Harmonic measure. The sorted list is given in SI1-D, and the top ranked peptides are those having relatively small similarity paths to all other nodes in the CSN.

This workflow allows the extraction of the most representative nodes/peptides describing the biologically-active chemical space (SI1-D). This representative subset can be used for multi-query similarity searches against peptide databases to retrieve all possible hits (Figure 9). The multi-query similarity search consists in using both the most central/representative nodes of the network communities and also the so-called singletons (isolated peptide nodes) as references/queries to retrieve the most similar peptides from databases by using local alignments. The best matches against the reference/query chemical space are determined by the maximum fusion rule by firstly ranking-down the similarity scores, to retrieve the best match between a query peptide and a target database and afterwards the best similarity scores are ranked for all reference peptides. Some studies have demonstrated that fusion by similarity scores and the maximum fusion rule are the best parameters for these models [126,127].

The integrated collection of 45 120 bioactive peptides registered in StarPepDB (http://mobiosd-hub.com/starpep/; accessed on 17 March 2022), that probably is the largest and most diverse bioactive peptide database to date, can be used for the discovering of central peptide nodes targeting an specific biological activity in the Chemical Space Networks (CSNs) and for taking advantage of them in multi-query similarity searches [33]. In this sense, Marrero-Ponce et al. explored different similarity networks of antiparasitic peptides (APPs) from StarPepDB to identify the most relevant and non-redundant APPs, that were later used as queries in similarity-based searches to identify potential APPs among non-labelled peptides as such in the StarPepDB. The proposed multi-query similarity search strategy outperformed state-of-the-art machine learning models aimed at APPs prediction like the AMPDiscover (https://biocom-ampdiscover.cicese.mx; accessed on 17 March 2022) and the AMPFun (http://fdblab.csie.ncu.edu.tw/AMPfun/index.html; accessed on 17 March 2022) webservers [30,31]. The methodology will also permit the design of new APPs by using the motifs found among the repurposed APPs [49]. More recently, a similar workflow using CSNs was applied to identify the most relevant tumor-homing peptides (THPs) within the StarPepDB. Such THPs were considered as queries (Qs) for multi-query similarity searches that apply a group fusion (MAX-SIM rule) model. The resulting similarity searching models outperformed state-of-the-art tools for THPs detection, and the best one was applied to repurpose AMPs from the StarPepDB as THPs. Novel THP leads were identified as well as new motifs accounting for their TH activity [50].

## 5. Models of Sequence Evolution for the Design and Optimization of Bioactive Peptides

Several *in silico* computational approaches inspired in molecular evolution events have been applied to the design and optimization of a peptide with a promising biological activity, known in medicinal chemistry as a “leading compound”. These algorithms are aimed to produce offspring peptides from a parent (hit peptide) until the “desired property” is meet according to selection criteria conducted either by ML prediction models or by biological assays (Figure 2). The offspring generation process can be iterated until reaching optimized peptidic scaffolds showing a trade-off between desirable/undesirable activities. The simulation process for generating offspring have evolved from inducing random mutations within the peptide sequence until guiding such aa substitutions under directed evolution concepts [41,128,129]. Although, algorithms inducing random mutations are commonly applied to generate sequence diversity in the peptide library, they could render unpredictable results that should be carefully analysed with selection algorithms. By contrast, computational algorithms inspired on directed mutagenesis have focused the design and optimization of “leading peptides” by guiding the generation of peptide offspring incorporating secondary structure features that influence positively on the antimicrobial activity such as amphipathic helices, kinked amphipathic helices, and other structures aimed to interact with lipid membranes [130].

Schneider et al. were the pioneers to apply simulated molecular evolution (SME) algorithms as a strategy for a rational peptide design by coupling the *in silico* generation of peptidases cleavage sites of 12 residues long to a selection mechanism represented by trained ANN [131,132]. The design was oriented to this region by generating offspring from a 12-residue sequence/peptide (parent sequence) which was iteratively mutated until meeting the best ANN quality classification metrics, used as a selection criterion of the design. The offspring sequence simulation was performed by introducing random mutations according to Gaussian-distributed probability values around the parent sequence. The mutation degree (small or large) is then conditioned by the estimation of position-specific mutability and the selected aa distance matrix [131,132]. As the position-specific mutability is averaged resulting the same for every position in the sequence; the aa mutation degree is determined by the aa substitution/scoring matrix type such the Grantham matrix [133], the Myata matrix [134], and the Risler matrix [135].

This SME approach was later applied by the same group to the optimization of anticancer peptides (ACP) aimed at improving their membranolytic activity and cell-type selectivity [51,136]. In [51], a known α-helical ACP served as the parent sequence for the generation of the offspring (ACP-derivatives). So that the generated offspring peptides retained similarity with the initial structural/property space and thus enabling a systematic optimization; the mutation function was controlled. This time the SME approach was accompanied with experimental measurements as a selection criteria or fitness objective within the optimization scheme. They used the half-effective concentration (EC_50_) on the breast cancer cell line MCF7 and the secondary structure preferences by circular dichroism (CD) spectroscopy as experimental filters. A similar SME protocol was applied in [136] to optimize the cell-type selectivity of the highest-scored candidate toward non-cancer cells and human erythrocytes. This candidate termed AmphiArc2 peptide resulted from the screening of virtual libraries generated by more advanced algorithms incorporating secondary structures features (alpha and amphipathic helices) that influence positively on the membranolytic action [130]. AmphiArc2 was selected as a parent sequence in the SME algorithm in which the mutated sequences (offspring) are generated from it. The offspring was scored according to a fitness function, defined by the anticancer activity and selectivity with respect to non-transformed cells. The best offspring was selected as a parent for the following optimization iteration [136].

Although the SME approach and the generation of oriented libraries toward certain secondary structures, relevant for the interaction with lipid membranes, have represented a step forward in the design and optimization pipeline of AMPs and ACPs [130,136], there still room for improving the simulation of molecular evolution of the offspring peptides. In this sense, algorithms that traditionally have been used for simulating sequence evolution in the field of molecular phylogenetics were recently applied to provide more rationality to the peptide library generation [52]. These algorithms were initially developed to evaluate the accuracy of MSA and phylogenetic reconstruction tools by generating sets of related simulated protein sequences from known phylogenies. The most representative ones are: ROSE (Random Model of Sequence Evolution) [137], SIMPROT (Simulation Protein Evolution) [138], and INDELible (Insertions and Deletions Simulator) [139]. In general, they are controlled by several evolutionary parameters such as tree topology, evolutionary distance matrices, mutation rate, insertion and deletion probabilities to simulate the evolution of offspring from a parent sequence. Ruiz-Blanco et al. incorporated the ROSE algorithm into the *de novo* design pipeline of peptide inhibitors of *E. coli* ATP synthase [52,76]. As parent peptides, both the natural inhibitor (IF_1_) of the mitochondrial ATP synthase and fragments of interfaces involved in protein—protein interactions between subunits of *E. coli* ATP synthase, were selected to generate peptide libraries. The residue conservation degree on these parent peptides was identified by MSAs within each class. A consensus parent peptide with its corresponding conservation scoring profile was estimated so different mutation rates to each position in the sequence could be assigned. This mutation probability vector together with a user-defined phylogenetic tree with a known topology and branch lengths guided the probabilistic function performing mutations, insertion and deletions on the parent peptide [52,76]. On the other hand, the sequence diversity of the offspring peptides in the library can be controlled by calibrating ROSE parameters against the pairwise identity [81]. A predefined binary phylogenetic tree with 1023 nodes and depth 9 implemented in ROSE was used in [52,76] for the generation of diversity-oriented libraries. The Figure 10 shows a schematic description of the ROSE algorithm.

Peptide libraries were screened by the PPI-Detect [77], an SVM-based model that predicts peptide interactions with both domains of the *E. coli* and human ATP synthases. As selection criterion, the high-scored interacting peptides with the *E. coli* ATP synthase but showing low values with the human’s were subsequently evaluated by in vitro inhibition tests. At applying advanced SME algorithms involving more evolutionary models/parameters like ROSE makes easier subsequently screening steps to find lead peptides at high success rate

## 6. Considerations in the Workflow for the High-Throughput Discovering of Bioactive Peptides

### 6.1. Brief Comparisons between High-Throughput (HT) and Classical Methods

The classical approach for discovery of bioactive peptides has changed from analysing biological extracts/fluids to perform a wide-genomics and proteomics search. In this sense both next-generation sequencing (NGS) technologies and mass spectrometry (MS)–based proteomics combined with bioinformatic tools have provided suitable approaches for the large-scale identification of bioactive peptides outperforming the classical methods. These last ones usually include a purification step combined with bio-guided assays, which require higher amount of biomass from the subject organism. Although they can determine the biological activity of bioactive compounds relatively at high accuracy, are time-consuming and the yield of bioactive compounds is low as well as the coverage of the chemical space [140]. On the contrary, the HT analyses can be performed with around 1 cm^3^ or 0.5–1 g of fresh or preserved tissues, for genomic/transcriptomic or proteomic purposes, respectively [53,141,142]. Generally, the HT methods allow covering the whole picture for potential bioactive compounds much faster. Despite HT methods usually require of powerful computational resources, both NGS and MS-based proteomics are becoming cheaper and their corresponding workflows are continuously optimized within the discovery process as well, resulting in a long-term sustainable approach [143,144]. Moreover, HT OMICs technologies yield a big amount of free public data, allowing the decentralization of the knowledge for the biodiscovery process.

Hence, the integration of OMICs approaches is more recommendable than the classical ones at the early stage of bioactive peptide discovery. However, bioassays-guided methods are still valid and complementary at advanced phases of the research [140,145].

### 6.2. Optimized Workflow for the Large-Scale AMPs Discovery from Profiled/Unexplored Organisms

Despite the advances in the discovery of bioactive peptides, improved protocols are still needed to increase the accuracy in both their large-scale identification and functional characterization, which is a major challenge, nowadays. Figure 11 illustrates the overall steps for the HT bioactive peptide discovery from model and unexplored organisms.

In order to analyze OMICs data released by NGS and MS-based HT proteomics, several computational/bioinformatic tools and platforms have been developed. Among them, for the *de novo* genome/transcriptome assembly we can mention, i.e., MIRA [146], Spades [147], CAP3 [148], OASES [149] and the Trinity package [150] including the *de novo* assembler and the TransDecoder for ORFs prediction (https://github.com/TransDecoder/TransDecoder/releases; accessed on 17 March 2022). Other ALL-IN-ONE licensed software like the toolbox CLC Genomic Workbench (CLC Bio-Qiagen, Aarhus, Denmark) [151] and OMICsBox (BioBam Bioinformatics, Valencia, Spain) [152], have integrated several tools for the complete workflow, including the *de novo* assemblers, custom/online/cloud functional annotation options with Blast+ [153], eggNOG [154], KEGG [155], providing as well as a set of functional analyses and statistical tests (i.e., Gene Ontology, deferential expression analyses and enrichment).

Among the NGS analyses, the RNA-seq has gained relevance because it can explore the coding regions of the genome by assembling, annotating and comparing expression profiles of the resulting transcripts [141,156]. Since elucidating the transcriptome demands lower computational cost than whole genome, and also provide useful information, its number has increasingly growth in databases. In this sense, transcriptomes from the same or related species are translated, usually with the TransDecoder or Six-Frame Translations Tool (S-FTT) (https://github.com/iracooke/protk; accessed on 17 March 2022), then annotated, and thus considered as reference database for improving protein identification in proteomics analyses from a target organism [157]. These are the grounds of proteogenomic analyses where genomic, transcriptomic and proteomic data are combined to assist the discovery of peptides from MS–based proteomic data, especially if they are not present in protein databases such as UniprotKB and other related ones (i.e, Swiss-Prot, TrEMBL and UniRef), the protein section of NCBI, Mendeley and ProteomExchange consortium [158]. On the other hand, the proteomic data can also be used to confirm gene expression [159].

In general, the overall proteomic approach for the discovery of bioactive peptides includes the following steps: (*i*) protein digestion, (*ii*) peptide separation, (*iii*) peptide fragmentation and MS spectra acquisition, (*iv*) peptide identification using MS spectra database by similarity searches or by *de novo* sequencing. In this sense, steps (*i*) and (*ii*) are addressed by several sample preparation protocols which selection determine the best yields/results. Specifically, for bioactive peptide discovery, it is advisable the solid-phase-enhanced sample-preparation (SP3) protocol [165] since it reaches a wider coverage of peptides than the filter-aided sample preparation (FASP) [166]; moreover, is less complicated and faster than the in-solution digestion (ISD) [167].

Besides to protocol improvements in sample processing [161], there have been advances in the peptide identification step by applying several computational strategies that have also refined their bioactivity prediction [159]. In addition to use transcriptomic data to increase peptide detection accuracy, the inclusion of custom databases is being applied to characterize the part of the proteome that remains unannotated. In this sense, composite databases have been explored for a deeper proteomic characterization of the salivary glands from *Octupus vulgaris* looking for revealing underexplored bioactive peptides/toxins from previous studies [54,157,161]. The composite database comprised data from the UniProtKB, built from *de novo* transcriptome assembly of Anterior (ASGs) and Posterior Salivary Glands (PSGs), combined with those retrieved from all transcriptomes available from the cephalopods’ PSGs. In addition, a comprehensive non-redundant AMPs database [124] was also included to provide additional insights about bioactive compounds such as putative AMPs [54]. In a previous work the same AMP subset was also considered as custom database to characterize the Ascidian tunic proteome by shotgun proteomics [53]. The computational analysis of the raw data implied searches against the Uniprot database (Bacteria and Metazoan section) and the AMP database. The Ascidian tunic revealed the presence of AMPs from both eukaryotes and prokaryotes and the “Biosynthesis of antibiotics” pathway was among the most significant ones, which support this tissue as an interesting reservoir of bioactive peptides/toxins and its role on the interactions Ascidians and their associated organisms. The AMP subset integrated in these previous analyses was published by Aguilera-Mendoza and probably was the most comprehensive and non-redundant AMP database reported so far [124], that later was updated in the StarPepDB (http://mobiosd-hub.com/starpep/; accessed on 17 March 2022) [78], as mentioned above.

Other important handicaps in the workflow of proteogenomic analysis are the False Discover Rate generated at analysing large protein/peptide databases [168,169,170] and the probable loss of information represented by small size transcripts encoding protein fragments < 100 aas that could be discarded by the TransDecoder [54,157,161], the tool dedicated to identify candidate coding regions within transcripts generated by *de novo* RNA-Seq, and such small-sized fragments could account for bioactive peptides. In order to perform a wider proteome analysis looking for uncovered AMPs and peptide toxins in the PSGs of *O. vulgaris,* contigs discarded in previous proteogenomic analyses (<100 aas) were translated with the S-FTT and then included in the protein database [54]. To optimize further proteogenomic analyses (i.e., time of analyses, FDR), or peptide annotation, sequences redundancy should be reduced with the CD-HIT [164] since the S-FTT generates many peptides sharing high similarities that could affect the overall peptide identification when increasing the FDR [170].

Other filters within the computational pipeline to process proteomic data have been applied to refine the search of peptide toxins against both canonical and custom databases. For example, the search can be framed against those toxins/peptides having signal peptides, responsible for their transport and secretion. Signal peptides have shown to contain common features across all life kingdoms [171]. In addition, cysteine-rich secretory proteins (CRISPs), small toxins (<100 aas) commonly found within the secretions of animal venoms, can be extracted from protein databases, to enrich reference databases for increasing proteomic toxin peptides detection [172]. Besides, the custom protein/peptide database can also be screened with ML-based tools e.g., ToxClassifier, that enables simple and consistent discrimination of toxins from non-toxin sequences [162], allowing the discovery of novel toxin-like bioactive peptides. Moreover, the fast genome-wide prediction of AMPs, using the ampir R package [160] can be used in the pipeline to retrieve novel peptides with antimicrobial signatures from public nucleotide databases, *de novo* transcriptomes/genomes assemblies, or as a filtering step before using S-FTT. More recently, new tools for the creation of proteogenomic databases considering the translation of pseudogenes, long non-coding RNAs (lncRNAs) and other non-canonical coding transcripts produced by alternative splicing, have allowed the identification of a significant number of cryptic peptides that may show interesting biological activities [163].

## 7. Concluding Remarks

Protein features inspired on molecular descriptors from chemoinformatics have emerged as successful predictors for AMPs activities. Particularly, ProtDCal’s descriptors have been recently incorporated in two RF-based webservers (AMPDiscover and ABPFinder) targeting AMPs predictions as well as their specific activities and putative bacterial targets. Moreover, ProtDCal’s descriptors have been involved in the design of antibiotic peptides by predicting their interaction to druggable targets from *E. coli*.

Among the recent ML approaches, undoubtedly DNNs have been the algorithm of choice for AMPs prediction in emerging tools. However, recently it has been shown that deep learning models’ performance in AMP prediction is comparable to the one of classical ML algorithms being their use mostly advisable when the performance gains justify the associated computational cost.

Currently, the network science implemented in StarPep is being applied as one of the top emerging approaches, regardless of ML, to assist the search and design of bioactive peptides through the identification of lead peptides within the known chemical space. On the other hand, methodologies that simulate sequence evolution in the phylogenetics field have been repurposed to assist the optimization of such peptide leads by generating diversity-oriented libraries which are strictly controlled by evolutionary parameters.

New considerations in analysing genomic, transcriptomic and proteomic data for AMPs discovery from either profiled or underexplored organisms are being also applied. Several filtering steps have been proposed to reduce the FDR in AMPs detection when custom databases are included, but at the same time, to encompass the highest number and diversity of peptides as possible.

## 8. Future Research Directions

Despite a great diversity of peptide features (classical and non-classical) that has been used in AMPs prediction/design, most of those features are sequences- or property- based; however, the 3D structural information of AMPs has not been deeply exploited for such aims [173,174,175]. Although experimental determinate 3D structures of AMPs are used in minor proportion than their sequences, the 3D structure prediction tools are becoming more accessible and less computational demanding when considering new advances in both software and computer architectures [176,177]. These facts will ease the gradually inclusion of 3D structural features in the prediction models.

Another alternative for the inclusion of higher structural information in AMPs encoding is the use of artificial representations, which have been commonly used in comparative analyses of DNA and proteins and in QSAR-type modelling [81]. The integration of peptide features from heterogeneous sources e.g., from pairwise alignments and peptide sequences into the same classifier could be another outlook for improving the classification rates of AMPs. The main problem is to figure out a framework to integrate them (the resulting features, not the source methodologies) into ML models training. As a clue for future research directions, the alignment- based and -free similarity measures were successfully integrated for training bigdata ML-based classifiers for orthologs detection [79]. Bigdata solutions applied to the prediction/optimization of AMPs have not been explored yet in spite of the fact that the number of AMPs has grown in databases as well as the number of features/descriptors that can be derived from them. Bigdata platforms could be applied when performing virtual screening of millions of peptides, especially if they are described with computationally demanding structural descriptors. As previously mentioned, it would be advisable that future ML models for the AMP prediction could consider the natural imbalance ratio between AMPs and non-AMPs as well as the existing one among the AMPs activities. Moreover, the prediction of AMPs activities should be addressed with fuzzy-based models since they generally show overlapping activities which are not evenly-distributed within the AMP population [23]. Therefore, the resulting predictions for AMPs activities may be scored with probability values and not only treated as a binary value. On the other hand, for peptide leads optimization, the offspring generation step is crucial for the overall process. This step generally is carried out by evolutionary algorithms that introduce structural diversity among child peptides somewhat randomly. Although these AI-based algorithms have been continuously evolving to guide such diversity in order to gain optimization efficiency; there is still room for improvements in this direction. Thus, the algorithms commonly used in phylogenetics for simulating sequence evolution could provide more rationality to the generation of offspring peptides since they have been designed with more evolutionary parameters that can be strictly controlled [52,76].

Finally, StarPep is probably the most promising methodology regardless of ML approaches, that has been reported so far. The complex network theory implemented in this tool has provided a different outlook to address several steps in peptide drug discovery process. StarPep bears particular analysis tools that have not previously reported for peptides, such as (*i*) the chemical space analysis of AMP databases by similarity networks, (*ii*) the identification of the most representative and non-redundant subset of AMPs from the original chemical space, (*iii*) the mapping of unlabelled peptide datasets on similarity networks built with the representative AMPs (*iv*) the multi-query similarity searches using representative peptides against target databases. Consequently, StarPep is becoming in a competing tool to the existing ML-based methods since it has being giving clues of improved classification rates [49,50], and because of its great potentialities for the identification and optimization of new peptide leads from either in silico generated peptide libraries or released data by the Omics techniques (Figure 11).

The effort of StarPepDB developers to gather all AMP databases in a non-redundant database [124] has shown a direct impact for the AMPs prediction tools [31]. However, the annotation quality for the reported AMPs must still be improved as well as the information on their biological or molecular targets. It is urgent that AMPs activity evaluations can be harmonized under the same protocols to construct more reliable benchmark datasets for the accuracy sake of the computational analysis tools. The diverse computational methods available for AMPs discovery are a powerful tool for the accurate design of peptide drugs. The growing availability of 3D structural descriptors and scoring functions will allow developing more effective in silico peptide drug design technologies. The assembling of ML methods with peptide-protein docking and molecular dynamics seems to be an effective alternative as well [178]. If all these aspects were considered for the computational-assisted search/design of peptide drugs, the next-generation of AMP leads will be more valuable for developing therapeutic agents to face challenging health problems such as cancer, infectious diseases and more recently, COVID-19.

## Figures and Tables

**Figure 1 antibiotics-11-00936-f001:**
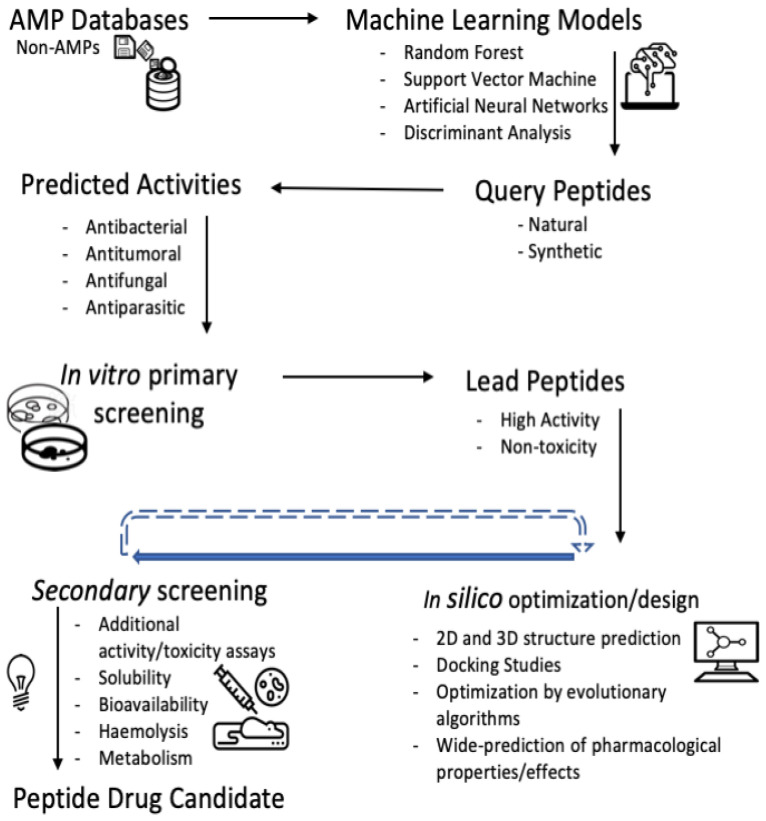
Workflow illustrating peptide drug discovery. The strategy involves the screening of query peptides from either natural or synthetic sources by applying ML models trained with the information stored in AMP databases. ML algorithms also assist the optimization/design step of lead peptides by means of a fitness/selection criterion [18,19].

**Figure 2 antibiotics-11-00936-f002:**
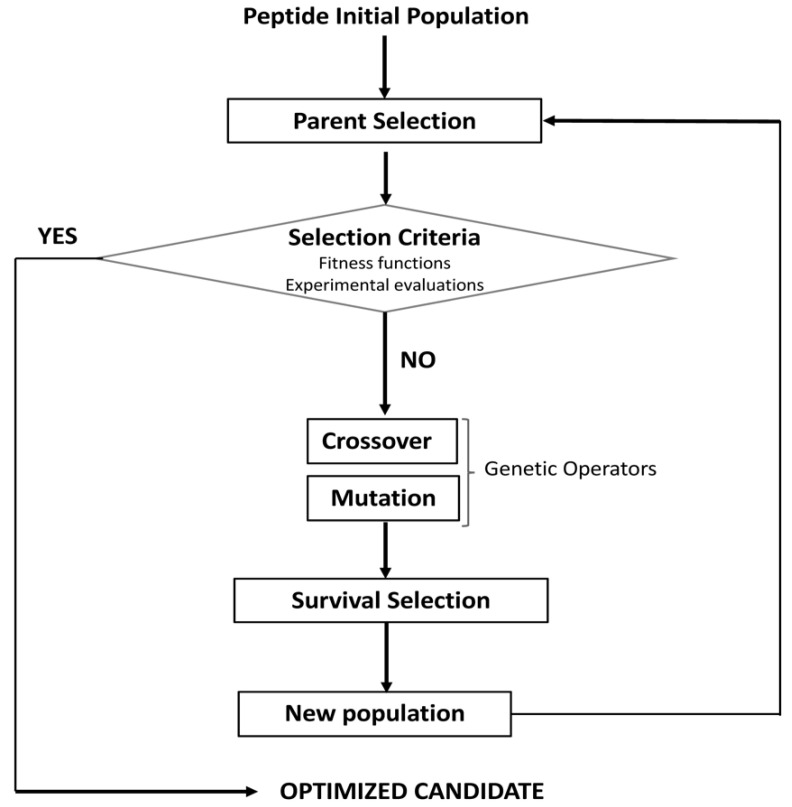
Workflow illustrating the main steps of evolutionary and genetic algorithms. Both approaches are very similar, in fact the use of evolutionary and genetic terms have been interchangeable. Genetic algorithms particularly use a fixed-length binary array to represent peptides as genes into a chromosome-like structure.

**Figure 3 antibiotics-11-00936-f003:**
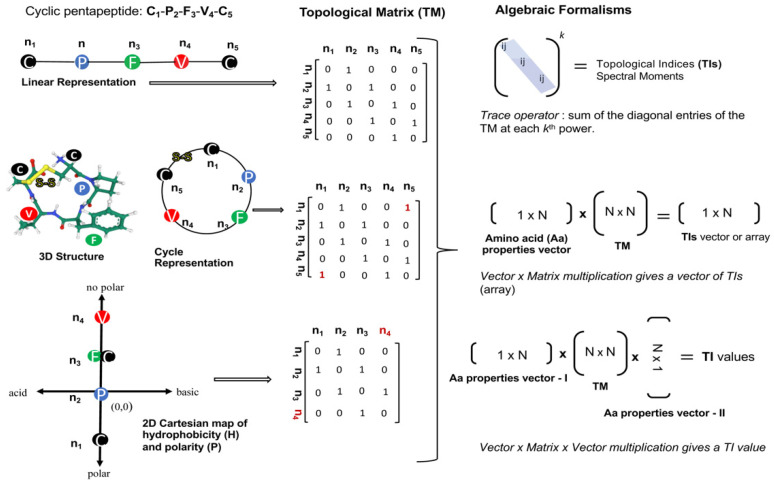
Workflow for the calculation of topological indices from several representation types of the cyclopentapeptide [CPFVC] with promising antiviral activity against the hantavirus cardiopulmonary syndrome [67]. Each peptide representation defines a singular topological matrix (TM) encoding structural features at different degrees. In addition to the several ways to represent the topology of a peptide (linear, circular, 2D-Cartesian), several algebraic formalisms/operators can be applied on the TM to calculate different topological indices (TIs) types. n represents the nodes in the peptidic representations (linear, circular, and Cartesian) as well as in their corresponding TMs, which may contain some elements in red font (e.g., n_4_ and 1) to highlight differences in structural encoding from the cyclopentapeptide. N indicates the number of rows and columns of matrices involved in TI calculation.

**Figure 4 antibiotics-11-00936-f004:**
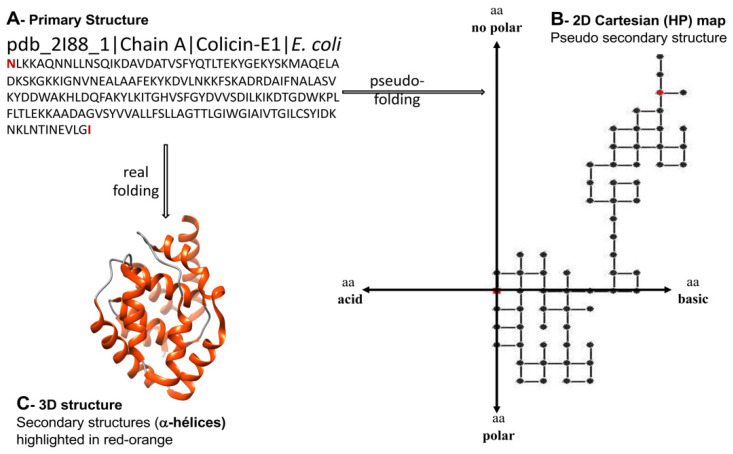
Different structural representations for the channel-forming domain of Colicin E1 (pdb 2I88). **A**—Primary structure, **B**—Pseudo secondary Cartesian map of hydrophobicity (H) and polarity (P) (2D Cartesian (HP) map), **C**—Three-dimensional structure. The 2D Cartesian protein map is an arbitrary bidimensional arrangement (pseudo-folding) of the protein/peptide sequences bearing higher-order useful patterns than contained in linear sequences.

**Figure 5 antibiotics-11-00936-f005:**
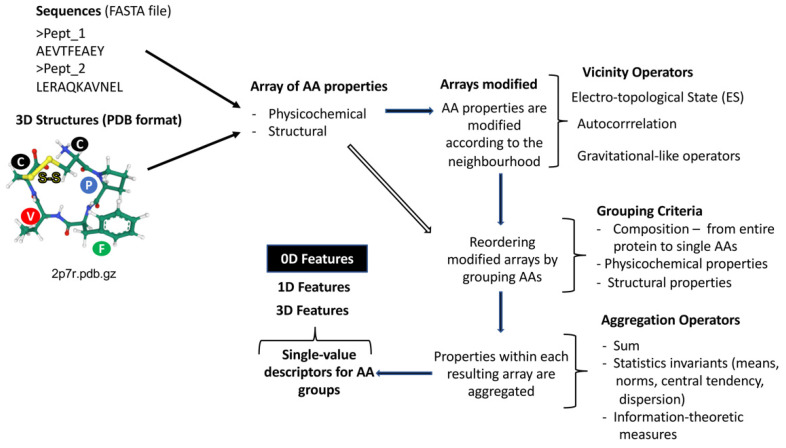
Schematic representation of ProtDCal’s descriptors calculation. 1D and 3D protein features implies the application of vicinity operators to modify amino acid (aa) properties while 0D features estimation go straightforward to group the original aa properties according to several grouping criteria.

**Figure 6 antibiotics-11-00936-f006:**
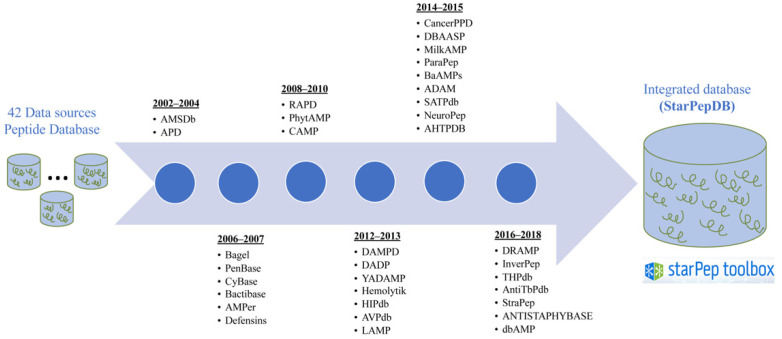
Chronological listing of AMP databases used in StarPep Database (StarPepDB) compilation. After collecting web pages from a large variety of bioactive peptide databases (see Table 1 in Ref. [78]), their contents were integrated into a graph database that holds total of 71.310 nodes and 348.505 relationships. In this graph structure, there are 45.120 nodes representing peptides (unique sequences) and the rest of the nodes are connected to peptides for the describing metadata.

**Figure 7 antibiotics-11-00936-f007:**
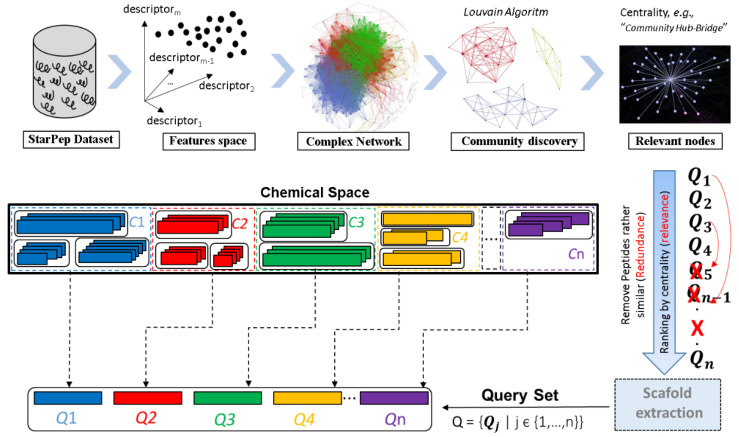
StarPep Toolbox flowchart. A flow diagram guiding the automatic construction and visual graph mining of similarity networks (see Figure 1 in Ref. [33]). Networks can be clustered, and communities are optimized using the Louvain method [125]. Moreover, the centrality of each node can be particularly measured by harmonic, community hub-bridge, betweenness, and weighted degree. Centrality is crucial to perform scaffold extractions because peptides are ranked according to their centrality score, and then redundant sequences are removed, prioritizing the most central. Thus, scaffold extractions depend on the type of centrality applied.

**Figure 8 antibiotics-11-00936-f008:**
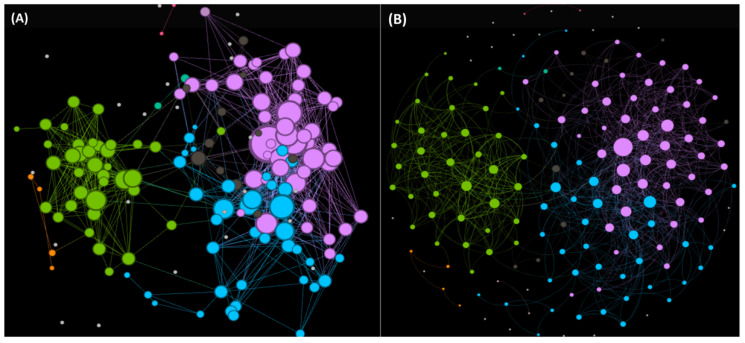
Visualizing the similarity network (Chemical Space Network, CSN) of a set of 174 non-redundant Anti-Biofilm Peptides (ABP_98% identity) at threshold t = 0.65 and density = 0.068, using the (**A**) three main PCAs as coordinated of each ABPs, and (**B**) Fruchtermann Reingold layout algorithm. Node colour represents the community (e.g., the biggest communities represented by cluster 3, 10 and 12 are in blue, purple and green colours, respectively), and node size symbolizes the centrality values. There are 20 ABP outliers (singletons). This figure has been created using the software starPep toolbox (version 0.8), available at http://mobiosd-hub.com/starpep; accessed on 17 March 2022.

**Figure 9 antibiotics-11-00936-f009:**
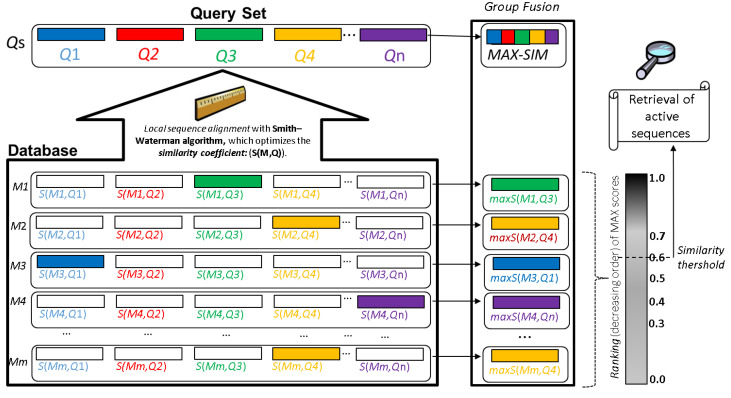
Schematic representation of the group fusion and similarity searching processes. Qi is a i peptide from a query/reference dataset, n is the number of peptides contained in a query dataset, S is identity coefficient between M and Q obtained by local alignment with Smith-Waterman algorithm, m is the number of peptides included in the target dataset. The similarity threshold is related to the percentage of identity.

**Figure 10 antibiotics-11-00936-f010:**
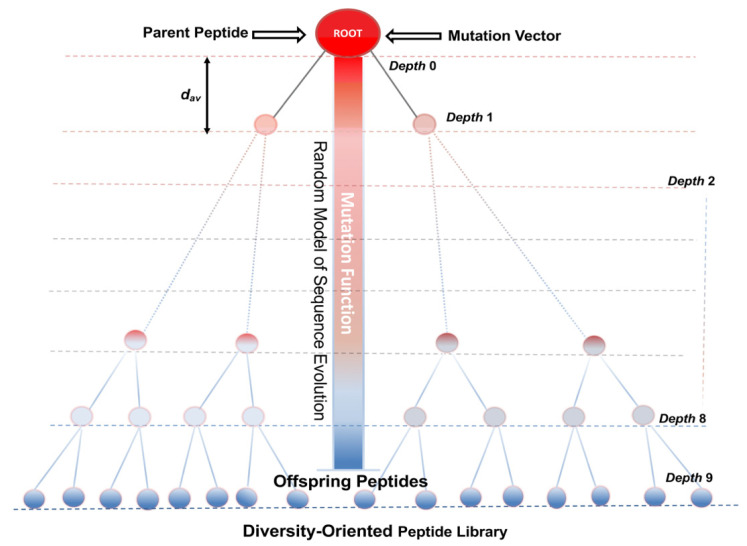
The binary mutation guide tree used by ROSE to mutate the parent/root peptide. The binary tree topology is determined by the number of nodes (1023), depth (9) and average distance (dav = 5–20 PAMs). Peptide library may be selected either from internal or terminal nodes of the tree. The identity percentage of the offspring peptides respect to the parent/root peptide is coloured-illustrated. Red colour means closely-related peptides to the parent while blue colour represents those distantly-related ones.

**Figure 11 antibiotics-11-00936-f011:**
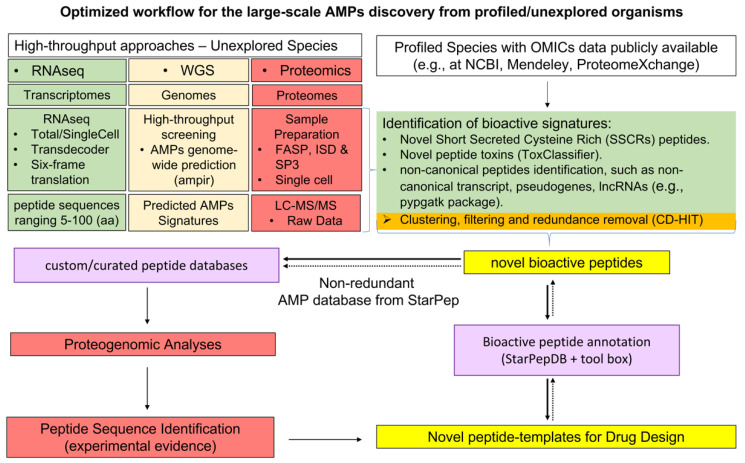
Optimized workflow for the high-throughput (HT) AMPs discovery from profiled and/or unexplored organisms. The figure summarizes the main phases in the AMPs discovery using genomic, transcriptomic and proteomic data from profiled or underexplored organisms. The figure depicts the pipeline for de novo HT discovery from un(der)explored organisms using OMICs approaches (shown in the top-left panel), and from nucleotide and proteomic information available at public databases (top-right). Genomic information publicly available at NCBI (Genome database https://www.ncbi.nlm.nih.gov/genome/; accessed on 17 March 2022) and transcripts encoding protein sequences under 100 aa length provided by the Transcriptome Shotgun Assembly (TSA) database (https://www.ncbi.nlm.nih.gov/genbank/tsa/; accessed on 17 March 2022), can be screened with the computational tool ampir for fast genome-wide prediction of AMPs [160]. Likewise, the remaining transcripts encoding peptides sequences ranging 5-100 aas length, usually discarded in transcriptomic analyses, can be translated with the six-frame translations tool (S-FTT) [157,161] after ORFs prediction with the TransDecoder. Considering bioactive peptides include animal toxins which are usually rich in cysteine, the aa sequences obtained with S-FFT can be either analyzed by the Proteomic toolkit (https://github.com/iracooke/protk; accessed on 17 March 2022) to identify cysteine-rich regions to discover novel Short Secreted Cysteine Rich (SSCRs) peptides, or by the Machine Learning (ML) tool ToxClassifier, that enables a simple and consistent discrimination of toxins from non-toxin sequences [162]. In addition, new tools like the pypgatk package [163] can recover a significant number of cryptic peptides of biomedical interest from pseudogenes, long non-coding RNAs (lncRNAs) and other non-canonical coding transcripts produced by alternative splicing. These filtering tools can be applied together CD-HIT [164] to screen nucleotide databases before custom and non-redundant peptide databases building for proteogenomic analyses or HT annotation. Finally, the StarPepDB with its associated tools [33] may have several roles within the presented workflow by providing non-redundant bioactive databases and also at reducing custom peptide databases with the identification of the most relevant peptides for proteogenomic analyses. Moreover, bioactive peptides detected in HT screening can be classified and clustered with StarPep in different categories according to their biomedical potential (e.g., AMPs, antitumor, antibacterial, antiparasitic, etc.).

**Table 1 antibiotics-11-00936-t001:** Summary of the most relevant ML approaches, from the classical to the emerging ones, for assisting the discovery of bioactive peptides from AMPs.

*Classical AMP Prediction Tools*				
**Integrated to Database**	**ML Algorithm**	**Peptide features**	**Implementation**	**Ref.**
CAMP_R3_	RF, SVM, ANN, DA	AAC, net charge, hydrophobicity	http://www.camp3.bicnirrh.res.in/prediction.php	[11]
DRAMP 3.0	ANN, SVM, RF	Secondary structure features	http://shicrazy.pythonanywhere.com/	[13]
ADAM	SVM	AAC	http://bioinformatics.cs.ntou.edu.tw/adam/tool.html	[14]
DBAASPv3.0	Threshold value-based discrimination	Physicochemical properties acconuting for the interaction with membrane	https://dbaasp.org/tools?page=general-prediction	[15]
**Independent Tools**				
ClasssAMP *****	RF, SVM	Sequence-based features	http://www.bicnirrh.res.in/classamp/predict.php	[35]
iAMPpred *****	SVM	compositional, physicochemical, and structural features	http://cabgrid.res.in:8080/amppred/server.php	[16]
iAMP-2L ******	k-NN	PseAAC	http://www.jci-bioinfo.cn/iAMP-2L	[25]
AmPEP	RF	Sequence-based features	https://cbbio.online/software/AmPEP/	[36]
amPEPpy	RF	Global protein sequence descriptors	https://github.com/tlawrence3/amPEPpy	[37]
AMPScannerv1 **	RF	Physicochemical features	https://www.dveltri.com/ascan/v1/index.html	[38]
AMPfun **	RF	AAC-based features, physicochemical features and word frequency-based features	http://fdblab.csie.ncu.edu.tw/AMPfun/index.html	[30]
AMAP **	SVM and XGboost tree	AAC-based features	http://amap.pythonanywhere.com/	[28]
*Emerging AMP prediction tools*				
MLAMP **	RF	Non-classical PSeAAC	http://www.jci-bioinfo.cn/MLAMP	[27]
IAMPE	RF, k-NN, SVM, XGboost	NMR-based features	http://cbb1.ut.ac.ir/AMPClassifier/Index	[34]
AMPDiscover **	RF/DNN	Non-classical protein features (ProtDCal)	https://biocom-ampdiscover.cicese.mx/	[31,39]
ABP-Finder **	RF	Non-classical protein features (ProtDCal)	https://protdcal.zmb.uni-due.de/ABP-Finder/index.php	[Unpub]
AMPScannerv2	DNN	AA alphabet	https://www.dveltri.com/ascan/v2/ascan.html	[19]
ACP-DL	DNN	Binary profile feature and K-mer sparce matrix	https://github.com/haichengyi/ACP-DL (Standalone)	[20]
xDeep-AcPEP *	DNN	Physicochemical, biochemical, evolutionary and positional	https://app.cbbio.online/acpep/home	[21]

Methods listed in Table 1 are currently active (Accessed on 7 March 2022) * Multi-label classifiers allowing the prediction of specific biological activities (antibacterial, antifungal, antiviral, antitumoral and others) from AMPs ** Hierarchical multi-label classifiers addressing firstly AMPs detection and in the second level their specific biological activities. ACC: amino acid composition, ANN: artificial neural networks, DA: discriminant analysis, DNN: deep neural networks, k-NN: k- nearest neighbours, NMR: nuclear magnetic resonance, PseAAC: pseudo amino acid composition, RF: random forest, SVM: support vector machine.

## Data Availability

Not applicable.

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
