# Peer review of "Emerging Computational Approaches for Antimicrobial Peptide Discovery"

_antibiotics, 2022, doi:10.3390/antibiotics11070936_

Round 1

Reviewer 1 Report

The authors explained and compared the existing methods very well. The review is very informative. However, the authors’ perspectives are missing from the manuscript. An additional section should be added to explain what novel questions should be addressed in order to increase the impact of future computation approaches in antimicrobial peptide discovery. Since the authors provided a comprehensive description of exciting methods, it should not be difficult to provide some ideas for future research in the field. 

Minor:

Figure1: Please provide the references in the caption from where this information is collected. 

Figure 2: This figure should be improved.

Figure 3: Each Symbol should be explained in the figure caption. for example, why n4 is in red in the third matrix.

Figure 10: I see more than two font sizes in the figure. Please improve this figure.

Author Response

Reviewer 1

Comments and Suggestions for Authors

The authors explained and compared the existing methods very well. The review is very informative. However, the authors’ perspectives are missing from the manuscript. An additional section should be added to explain what novel questions should be addressed in order to increase the impact of future computation approaches in antimicrobial peptide discovery. Since the authors provided a comprehensive description of exciting methods, it should not be difficult to provide some ideas for future research in the field. 

Many thanks for your useful comments. A new section entitled “Future Research Directions” was included at the end of the manuscript.

Minor:

  1. Figure1: Please provide the references in the caption from where this information is collected. 

References 17 and 18 were inserted in the Figure 1 caption. They were used for the figure construction.

  1. Figure 2: This figure should be improved.

A new version of figure 2 was inserted, and the text related to it was also updated in the manuscript.

  1. Figure 3: Each Symbol should be explained in the figure caption. for example, why n4 is in red in the third matrix.

The following sentences were added to the Figure 3 caption: “n represents the nodes in the peptidic representations (linear, circular and cartesian) as well as in their corresponding TMs which may contain some elements in red font (e.g. n4 and 1) to highlight differences in structural encoding from the cyclopentapeptide. N indicates the number of rows and columns of matrices involved in TIs calculation”

  1. Figure 10: I see more than two font sizes in the figure. Please improve this figure.

Figure 10 was improved. A new version was inserted in manuscript

Reviewer 2 Report

1.      The rise of resistance to antimicrobial agents evidenced in the last decades have caused excess healthcare costs worldwide. Reference?

2.      The table 1 displays the most relevant ML approaches, from the classical to the emerging ones, for assisting the discovery of bioactive peptides from AMPs. Rewrite.

3.      Future perspectives highlighting the present shortcomings that can be explored in coming times must be mentioned.

4.      The review provides in-depth knowledge about the concerned area but the information seems much more than it should be present. It can be made crisper removing some unnecessary details from the manuscript.

5.      Check for erratic usage of abbreviations and use of long sentences.

Author Response

Reviewer 2

Comments and Suggestions for Authors

  1. The rise of resistance to antimicrobial agents evidenced in the last decades have caused excess healthcare costs worldwide. Reference?

The following new reference was inserted “Antimicrobial Resistance, C. Global burden of bacterial antimicrobial resistance in 2019: a systematic analysis. Lancet 2022, 399, 629-655, doi:10.1016/S0140-6736(21)02724-0”

  1. The table 1 displays the most relevant ML approaches, from the classical to the emerging ones, for assisting the discovery of bioactive peptides from AMPs. Rewrite.

It was corrected according to the reviewer suggestion

  1. Future perspectives highlighting the present shortcomings that can be explored in coming times must be mentioned.

A new section entitled “Future Research Directions” was included at the end of the manuscript

  1. The review provides in-depth knowledge about the concerned area but the information seems much more than it should be present. It can be made crisper removing some unnecessary details from the manuscript.

We have condensed some parts in the manuscript (see subsections 2.1.1, 6.1 and 6.2) intending to remove unnecessary details as the reviewer suggested.

  1. Check for erratic usage of abbreviations and use of long sentences.

The writing and grammar were revised again along the text.

Reviewer 3 Report

General Comments

Reviewed is the manuscript “Emerging Computational Approaches for Antimicrobial Peptide Discovery” submitted by Guillermin Aguero-Chapin, et, al. This manuscript contains a very comprehensive review of the computational approaches for antimicrobial discovery in recent years. The article is well organized with a smooth flow of information during the explanation of each method. It is well-written, with very few clerical errors, and the style and layout are very well articulated. Overall, the authors clearly demonstrate their approach and detail the performance gained in this research field and the article meets the required standards for publication after minor edits.

 Specific Comments:

-      Within the discussion/conclusion, the authors emphasized the pros of ML approaches. However, additional discussions are recommended to balance different techniques.

-      In table 1, some of the tools’ links are no longer working during the peer review process. (For instance, http://www.jci-bioinfo.cn/iAMP-2L; https://omictools.com/ampep-tool) Please check and clarify, and consider updating/removing the outdated tools or links throughout the whole manuscript.

-      Review the written portion of the thesis for grammatical errors, especially the subject-verb agreements.

Author Response

Reviewer 3

General Comments

Reviewed is the manuscript “Emerging Computational Approaches for Antimicrobial Peptide Discovery” submitted by Guillermin Agüero-Chapin, et, al. This manuscript contains a very comprehensive review of the computational approaches for antimicrobial discovery in recent years. The article is well organized with a smooth flow of information during the explanation of each method. It is well-written, with very few clerical errors, and the style and layout are very well articulated. Overall, the authors clearly demonstrate their approach and detail the performance gained in this research field and the article meets the required standards for publication after minor edits.

 Specific Comments:

  1. Within the discussion/conclusion, the authors emphasized the pros of ML approaches. However, additional discussions are recommended to balance different techniques.

Although we have discussed each methodological issue in their corresponding sections/subsections, a new subsection entitled “Future Research Directions” was included at the of the manuscript where debates on other topics than ML were added.

  1. In table 1, some of the tools’ links are no longer working during the peer review process. (For instance, http://www.jci-bioinfo.cn/iAMP-2L; https://omictools.com/ampep-tool) Please check and clarify, and consider updating/removing the outdated tools or links throughout the whole manuscript.

Thanks for your comments. We have updated the link related to the AmPEP tool and revised the links related to all tools referenced in the manuscript. Although the iAMP-2L currently is not currently active (don´t know the reason why?) but at the moment of the revision was active (March – 2022), that`s why was included in Table 1. iAMP-2L is a very popular method that have been used for many researchers to predict/compare the AMP activities. So, we would like to let the tool in the manuscript by specifying the access date in the Table footnote.

  1. Review the written portion of the thesis for grammatical errors, especially the subject-verb agreements.

The grammar was revised again along the text.

Round 2

Reviewer 1 Report

The authors incorporated previous comments carefully. I do not have any further comments.